# VAPB ER-Aggregates, A Possible New Biomarker in ALS Pathology

**DOI:** 10.3390/cells9010164

**Published:** 2020-01-09

**Authors:** Maria Piera L Cadoni, Maria Luigia Biggio, Giannina Arru, Giannina Secchi, Nicola Orrù, Maria Grazia Clemente, GianPietro Sechi, Alfred Yamoah, Priyanka Tripathi, Sandro Orrù, Roberto Manetti, Grazia Galleri

**Affiliations:** 1Department of Medical Surgical and Experimental Sciences, University of Sassari, Viale San Pietro 8, 07100 Sassari, Italy; mariapieracadoni@libero.it (M.P.L.C.); maribi84@gmjail.com (M.L.B.); parentina@yahoo.com (G.A.); gsecchi@uniss.it (G.S.); mgclemente@uniss.it (M.G.C.); rmanetti@uniss.it (R.M.); 2Department of Medical Sciences and Public Health, University of Cagliari, S. P. Monserrato—Sestu km 0,700, 09042 Monserrato, Italy; nicola.orru@atssardegna.it (N.O.); s.orru@unica.it (S.O.); 3Institute of Neuropathology, RWTH Aachen University Medical School, Pauwelsstr. 30, 52074 Aachen, Germany; ayamoah@ukaachen.de (A.Y.); ptripathi@ukaachen.de (P.T.)

**Keywords:** ALS, biomarker, VAPB ER-aggregates, endoplasmic reticulum, flow cytometry, PBMC, immunofluorescence

## Abstract

A point mutation (P56S) in the gene-encoding vesicle-associated membrane-protein-associated protein B (VAPB) leads to an autosomal-dominant form of amyotrophic lateral sclerosis (ALS), classified as ALS-8. The mutant VAPB is characterized by ER-associated aggregates that lead to a complete reorganization of ER structures. Growing evidences suggest VAPB involvement in ALS pathomechanisms. In fact, numerous studies demonstrated VAPB alteration also in sporadic ALS (sALS) and showed the presence of its aggregates when others ALS-related gene are mutant. Recently, the identification of new biomarkers in peripheral blood mononuclear cells (PBMCs) has been proposed as a good noninvasive option for studying ALS. Here, we evaluated VAPB as a possible ALS pathologic marker analyzing PBMCs of sALS patients. Immunofluorescence analysis (IFA) showed a peculiar pattern of VAPB aggregates in sALS, not evident in healthy control (HC) subjects and in Parkinson’s disease (PD) PBMCs. This specific pattern led us to suppose that VAPB could be misfolded in sALS. The data indirectly confirmed by flow cytometry assay (FCA) showed a reduction of VAPB fluorescent signals in sALS. However, our observations were not associated with the presence of a genetic mutation or altered gene expression of VAPB. Our study brings further evidences of the VAPB role in ALS as a diagnostic biomarker.

## 1. Introduction

Amyotrophic lateral sclerosis (ALS) is a progressive neurodegenerative disease that affects both lower motor neurons in brainstem and spinal cord and upper motor neurons in motor cortex [1,2]. ALS pathogenesis is poorly understood, and currently, there is neither a cure nor a specific diagnostic test for this disease. Approximately 90% of ALS cases are sporadic (sALS), while the remaining 10% of cases are familial (fALS) [3,4]. Several genes involved in ALS pathomechanisms have been identified [5,6,7], and all of these can cause motor neurons death through different mechanisms such as oxidative stress, mitochondrial dysfunctions, protein aggregates formation, glutamate toxicity, and neuroinflammation [7,8,9,10]. Among ALS-causative genes, a point mutation, proline to serine substitution at position 56 (P56S), in a vesicle-associated membrane-protein-associated protein B (VAPB)-encoding gene, responsible for a rare fALS case and classified as ALS-8 [11], has been identified, which is associated with a highly variable clinical course [12]. VAPB belongs to the family of vesicle-associated membrane-protein-associated proteins, endoplasmic reticulum (ER) resident proteins [13,14], which play different roles in cells for regulation of calcium homeostasis, vesicle trafficking, bouton formation at a neuromuscular junction, microtubules organization, lipid transport, and unfolded protein response (UPR) [14,15,16,17]. The most frequent mutation in VAPB-encoding genes is P56S [11]. However, other more rare mutations have been identified [12]. The presence of VAPB mutant genes causes an impairment of the protein, resulting in intracellular protein inclusions, already characterized in transgenic mice [11], which entails an abnormal reorganization of an ER structure and alters cellular homeostasis [11,13,15,18,19,20,21,22]. Several studies in human and transgenic mice have documented that VAPB inclusions also occur when other ALS-causative genes are present, such as superoxide dismutase 1 (*SOD1*) [23], TAR DNA-binding protein (*TDP43*) [24], and chromosome 9 open reading frame 72 (*C9orf72*) repeat expansion [25]. Moreover, there are evidences that suggest VAPB involvement in sALS [11,15,22]. In the present paper, we studied VAPB as a possible ALS pathologic marker by analyzing peripheral blood mononuclear cells (PBMCs) isolated from sALS-affected patients. In fact, PBMCs share more than 80% of their transcriptomes with other tissues, including the central nervous system, and can be isolated easily from patients by venipuncture [26]. Previous studies demonstrated that PBMCs are a valid cell model to study ALS [26,27,28,29]. Based on this and on the new vision of ALS as a pathology affecting not only motor neurons exclusively but also other cell systems [30,31,32], we investigated the possible role of VAPB as a pathologic marker in PBMCs from patients with sALS.

Before performing our study on patients, we first reproduced the ALS-8 model transfecting HeLa cells with wild type (Wt) and VAPB-P56S to have a reference model already characterized. Next, we translated the study on sALS patients, using also PBMCs of Parkinson’s disease (PD) patients and healthy subjects as neurological controls and healthy controls (HCs), respectively. The data obtained by performing immunofluorescence analysis (IFA) revealed, in all sALA patients, the presence of VAPB ER-aggregates was consistent with that obtained in HeLa cells transfected with mutant VAPB. The flow cytometry assays (FCAs) in sALS PBMCs and HeLa cells revealed a statistically significant reduction in VAPB medium intensity of fluorescence (MFI) compared to that in the controls, suggestive of possible VAPB misfolding. However, all alterations in VAPB were not associated with genetic mutation or altered gene expression of the protein. Our study provides further evidence for a VAPB role in ALS. The data obtained are relevant in terms of ALS diagnosis and pathomechanisms.

## 2. Materials and Methods

### 2.1. HeLa Cells Transfection and Cultures

HeLa cells were transfected, when 60–70% confluence was achieved. Overexpression of enhanced green fluorescent protein (EGFP), VAPB-Wt, and VAPB-P56S was obtained using the Lipofectamine 2000 transfection reagent (0.70 Μl; Thermo Fisher Scientific, Waltham, MA, USA) in combination with an Opti-MEM (25 μL) medium according to the manufacturer’s protocol. Genomic DNA was diluted in Opti-MEM to achieve a final concentration of 0.01 μg/μL. A mixture (Lipofectamine, medium, and plasmid-DNA) was added, and cells were incubated at 37 °C for 4 h. Next, cells were cultured in a mixture of DMEM + 10% fetal bovine serum (FBS) at 37 °C, and successful transfections were verified by western blot. Plasmids used in this study were a kind gift from the Institute of Neuropathology Aachen, Germany; details of plasmid construction were described in an initial study performed [33]. Briefly, VAPB cDNA was subcloned in the pCDNA3.1/*myc*-HIS expression vector, and the point mutation P56S was introduced into a full-length DNA sequence of VAPB. These plasmids then were subcloned further in the EGFP-N1 vector.

### 2.2. Ethical Committee Approval

The institutional review board (Azienda Ospedaliera Universitaria, Sassari, Italy: N° 2395/2016) approved the study protocol; all participants provided the written informed consent.

### 2.3. Patients and Methods

The study cohort consisted of 24 sALS patients with definite diagnosis, 14 patients with PD, and 24 HCs (Table 1). Patients with sALS and PD were enrolled at the Neurological Clinic and Pneumological Clinic of the University Hospital of Sassari and the ATS (Azienda Tutela Salute) clinic of the local health district, specialized in home care for ventilated patients. The Blood Transfusion Centre of Sassari provided HC subjects. All methods were performed in accordance with relevant guidelines and standards.

### 2.4. PBMC Isolation

For each patient and each control subject, 20 mL of peripheral venous blood was collected in sodium heparin tube tests. A total volume of blood was mixed with an equal volume of Ficoll Histopaque^®^ (Sigma-Aldrich, St. Louis, MO, USA). After centrifugation at 300× *g* for 20 min, PBMCs ring was collected, washed twice in a phosphate-buffered saline (PBS) solution and centrifuged for 10 min at 300× *g*. The supernatant was discarded by inversion, and the PBMCs pellet was resuspended in PBS for cell counting in a Burker counting chamber (Sigma-Aldrich).

### 2.5. Western Blot Analysis

Transfected HeLa cells and PBMCs derived from patients with sALS and PD patients and HCs were resuspended in a lysis buffer containing 0.5% Triton X-100 in PBS (pH 7.4), 0.5 mM phenylmethylsulfonyl fluoride, and complete protease inhibitor cocktail. After incubation on ice for 30 min, lysates were briefly sonicated. Protein concentrations were determined according to the method of Bradford using a Bio-Rad protein assay reagent (Bio-Rad, Hercules, CA, USA). For each analysis, around 15 µg of protein was loaded and separated by sodium dodecyl sulfate-polyacrylamide gel-electrophoresis (SDS-PAGE) according to the Bio-Rad protocol. Afterwards, electrophoresis protein bands were transferred on a polyvinylidene difluoride (PVDF) membrane, followed by blocking with 4% skimmed milk to prevent nonspecific binding. Membranes then were incubated overnight with a primary antibody (ubiquitin (1:1000) from Agilent Technologies, Santa Clara, CA, USA; LC-3 (1:2000) and p62 (1:1000) from MBL International; Woburn, MA, USA, and GRP78 (1:1000) Cell Signaling Technology, Leiden, The Netherlands, HSP90 (1:1000), and β-tubulin (1:10,000) from Sigma-Aldrich). The next day, cells were washed three times with tris-buffered saline and 10% Tween 20 (TBS-T) and incubated for one hour at room temperature with a secondary antibody (anti-rabbit antibody (1:10,000) with a Pierce substrate; Invitrogen anti-mouse antibody (1:10,000)) conjugated to horseradish peroxidase (HRP). Proteins were visualized using “Super Signal Pico” and “Super Signal Fento” (Thermo Fisher Scientific, St. Louis, MA, USA). Chemiluminescent imaging was obtained using a ChemiDoc MP Imaging System (the University of North Carolina at Chapel Hill), and densitometry values were determined using the software ImageJ (the National Institutes of Health (NIH), Bethesda, MD, USA). Protein densitometry values for each sample were normalized with a β-tubulin value to obtain semiquantitative measures of protein expression. All western blot experiments were performed in triplicate.

### 2.6. Primary Cutaneous Fibroblasts Cells Cultures

Primary cutaneous fibroblasts from five patients with sALS, (two with an A382T mutation in TDP-43-encoding gene and the remaining three without mutations in the most common ALS-related genes (*SOD1*, *FUS*, *C9orf72*, and *TDP-43*) and sex and age-matched healthy donors were cultured in DMEM with 10% FBS under conditions of 5% CO_2_ and 37 °C until use. All these cells were kindly shared with us by our collaborator Prof. Sandro Orrù, the Department of Medical Genetics of Cagliari University, from a cell bank financed by the foundation AriSLA for “Progetto Eugenio”. 

### 2.7. Immunofluorescence

Cells were fixed with 4% paraformaldehyde (PFA) (Sigma-Aldrich) at room temperature and permeabilized with PBS/0.1% Triton X-100. Cells then were washed using PBS/1% bovine serum albumin (BSA) and stained separately with a primary antihuman VAPB monoclonal antibody (PBS/1% BSA (1:500); Abcam, Cambridge, the United Kingdom) and an antihuman VAPB custom-made rabbit polyclonal antibody (1:500) (gift from Dr. Goswami and Prof. Joachim Weis at the Institute of Neuropathology, Uniklinik, Aachen) [33,34,35]. Next, cells were washed in PBS/1% BSA and incubated with a secondary antibody (anti-mouse IgG, (1:500) PBS/1% BSA conjugated with fluorescein isothiocyanate (FITC) or FITC anti-rabbit IgG, (1:500); Sigma-Aldrich). Approximately 2 × 10^4^ cells were used for the staining, and cover glasses were mounted with 4’,6-diamidino-2-phenylindole (DAPI) (ProLong Gold antifade reagent; Molecular Probes, Eugene, Oregon, the United States). Fluorescent signals were acquired with a confocal microscope (Leica Microsystems Srl, Milan, Italy), and digital images were assembled using Adobe Photoshop. For each sample, 20 fields were analyzed., A ratio of the number of cells with VAPB aggregates to the original total number of cells used for the assay (2 × 10^4^) was determined for each subject.

Of note, a monoclonal antibody used for this assay (clone number: MM0949-33M9) recognizes the N-terminal domain of VAPB (aa 2-130) called major sperm protein (MSP) domain, which guides the correct folding of the protein [14] (Figure 1), while a polyclonal antibody was made using the N-terminal aminoacid sequence (NH2-VEQVLSLEPQHEC-CONH2) of human VAPB.

### 2.8. Flow Cytometry Assays

FCAs were performed in transfected HeLa cells and PBMCs. As for the IFA studies, around 2 × 10^6^ cells were fixed in 4% PFA, permealized with PBS/0.05% saponin and incubated for 30 min with either a custom-made primary polyclonal antihuman VAPB (1:500) or an antihuman VAPB monoclonal antibody (1:500; Abcam, Cambridge, the United Kingdom). After incubation, cells were washed with PBS/0.05% saponin and incubated in a secondary antibody (FITC anti-rabbit IgG (1:500) or FITC anti-mouse IgG (1:500); Sigma-Aldrich). Cells were then analyzed by a flow cytometer FACSCanto (Becton Dickinson, Franklin Lakes, NJ, USA). The flow cytometer was set to acquire 3 × 10^4^ total events. Data analysis was carried out using BD FACSDiva™ software 6.1.3. The fluorescence signal was expressed as a mean fluorescence intensity (MFI). 

### 2.9. VAPB mRNA Expression Gene

RNA was isolated from PBMCs using the TriPure isolation reagent (F. Hoffmann-La Roche AG, Basel, Switzerland) and the PureLink RNA Mini Kit ((Thermo Fisher Scientific) following the manufacturers’ protocols. The concentration and the quality of RNA were evaluated with the Thermo Scientific ™ NanoDrop 2000 spectrophotometer (Thermo Fisher Scientific). Total RNA was reverse-transcribed into cDNA using Maxima reverse transcriptase (Thermo Fisher Scientific) and analyzed by (q)PCR analysis using SYBR Green mix (Kapa Biosystems, Wilmington, MA, USA) and gene-specific primers (*forward 5′-3′ TGTAAGAGGCTGCAAGGTGA*; *reverse 5′-3′ ATGCTGAAATGGGGCTGTTG*). Relative mRNA expression levels were calculated by the 2**^−ΔCt^** method using glyceraldehyde 3-phosphate dehydrogenase (GAPDH) as an internal control.

### 2.10. Genomic Analysis

Genomic DNA from patients and controls was extracted using the QIAamp DNA Mini Kit (QIAGEN S.r.l., Milan, Italy). Each DNA sample were resuspended and quantified by a fluorimetric method (Qubit 4 Fluorometer; Thermo Fisher Scientific). Genetic analysis was performed by studying the major genes known to be involved in ALS pathogenesis: *SOD1*, *FU*S, *TARDBP*, *C9Orf72*, and VAPB-encoding gene. 

Specific primers for long-range (LR) PCR were designed to amplify all parts of the genes *SOD1*, *FUS*, *TARDBP*, and VAPB-encoding genes (Table 2). Primers were designed using Primer-BLAST [36]. LR PCR was carried out using PrimeSTAR^®^ GXL DNA Polymerase kit (Takara, Shiga, Japan) and then 50 ng of genomic DNA. LR PCR products were used to prepare libraries for next-generation sequencing (NGS) assays using Nextera DNA Library Prep Kit (Illumina, San Diego, California, the United States). Libraries were normalized at a 3–4 nM fragment concentration, and denaturation in 0.1 N NaOH were sequenced in a MiSeq instrument (Illumina, San Diego, California, the United States). Five hundred cycles of sequence were applied for the sequencing run. Variants were found using the MiSeq Reporter software package v2.6 and annotated by the software Illumina Variant Studio 3. The characterization of a hexanucleotide-repeated region of the *C9orf72* gene was obtained utilizing a protocol described by De Jesus et al. (2011) [37]. The repeating number was quantified by a fragment analysis obtained with capillary electrophoresis in an ABI PRISM^®^ 3500 sequencer and the software GeneMapper (Thermo Fisher Scientific).

### 2.11. Statistical Analysis

All data obtained were analyzed using GraphPad Prism 6.0 software (GraphPad Software, San Diego, CA, USA). Continuous variables were presented as mean ± standard deviation (SD) and categorical variables were presented as numbers and percentages. A parametric Student’s t-test was used to compare two groups, and a value of *p* < 0.05 was considered significant. 

## 3. Results

### 3.1. Cellular Model Analysis

In HeLa cells stained by IFA with an antihuman VAPB polyclonal antibody, VAPB ER-aggregates were present in those transfected with mutant VAPB (P56S), but not in VAPB-Wt HeLa cells (Figure 2a). As others have shown, these aggregates are caused by VAPB misfolding [13,16,18,21] that lead to ER reorganization [11,13,38]. Performing the FCA with the antihuman VAPB monoclonal antibody, we observed a reduction in VAPB fluorescent signal in VAPB-P56S-transfected HeLa cells compared to in VAPB-Wt Hela cells (*p* = 0.0003). This evidence revealed that, in presence of VAPB misfolding, the monoclonal antibody lost the ability to bind efficiently its recognition site to the protein, causing a reduction of fluorescent signal detection (Figure 2b). The western blot analysis confirmed the data reported in the literature. In fact, as expected, VAPB-P56S cells showed increased levels of GRP-78, HSP-70, P62 LC3 I and II, and ubiquitin (Figure 2c,d), all cellular stress markers that are overexpressed in presence of misfolded protein accumulation [39,40,41,42,43,44,45].

### 3.2. VAPB ER-Aggregates in sALS Patients PBMCs

The pattern of VAPB immunofluorescence was substantially different in PBMCs from sALS patients compared to those obtained from patients with PD and HCs (Figure 3). The stain with an antihuman VAPB polyclonal antibody revealed a uniform signal in HCs (Figure 3a–c) and patients with PD (Figure 3d–f) PBMC cytoplasm, while fluorescence patterns in all the sALS patients PBMCs were characterized by numerous VAPB clusters distributed around the nucleus (Figure 3g–i). These VAPB aggregates were similar to the staining pattern of mutant VAPB-P56S overexpressed in HeLa cells (Figure 2a) and might be a representative of defective ER organization [4,11,13,21] caused by VAPB misfolding. An average of 70% of total PBMCs carrying ER-aggregates analyzed were observed in sALS subjects compared to an average of 2% in PD patients (*p* < 0.00001) and 0% in HCs (*p* < 0.00001). The monoclonal antibody showed a similar pattern compared to the polyclonal antibody, but its signals were basically weaker in HCs and PD patients and were basically negative in all the sALS (data not shown).

### 3.3. VAPB Accumulations in sALS Patients Fibroblasts

Prompted by the observation of VAPB accumulations in PBMCs of the sALS patients, we asked if we could detect similar early changes of VAPB in fibroblasts obtained from skin biopsy of sALS patients. Applying the same antihuman VAPB polyclonal antibody on the sALS and healthy donor fibrosblasts, we observed, in sALS, a globular accumulation of VAPB (Figure 4a), which were consistent with the data obtained from PBMC analysis. VAPB aggregates were colocalized with GRP78, a trasmembrane protein of ER (Figure 4b), confirming that these aggregates are strongly associated to this cellular organel.

### 3.4. Decrease of VAPB Fluorescent Signals in sALS Patients 

The findings of VAPB aggregates in PBMCs and fibroblasts from sALS patients obtained by IFA prompted us to suppose that VAPB could be misfolded in sALS. Therefore, we quantified, by an FCA, fluorescent signals of VAPB protein using antihuman VAPB polyclonal and monoclonal antibodies. The data obtained by a monoclonal antibody assay confirmed indirectly our hypothesis of VAPB misfolding. In fact, we revealed a statistically significant reduction of VAPB fluorescent signals (MFI) in sALS compared to in HCs with a *p*-value of 0.018 and PD controls with a *p*-value of 0.003 (Figure 5). This result revealed a reduced ability of the monoclonal antibody to bind its recognition site (see Figure 1) only in sALS, supposing that this protein site was misfolded or hidden. On the contrary, the same analysis performed with the polyclonal antibody did not reveal significant differences of VAPB fluorescent signals between patients with sALS and PD and HCs (Appendix A), most likely due to the fact that the polyclonal antibody recognized more VAPB epitopes than the monoclonal antibody.

### 3.5. Nonalteration of VAPB mRNA Expression in sALS Patients 

On the basis of the results obtained by cytometric analysis, we decided to quantify VAPB gene expression by real-time PCR. In the current literature about VAPB expression, some authors declare that both VAPB and VAPA are reduced in human ALS patients and SOD1 ALS transgenic mice, suggesting that VAP family proteins may be involved in the pathogenesis of sporadic and SOD1-linked ALS [13]. However, the data obtained revealed the expression of VAPB mRNA in sALS patients (*n* = 24; mean ± SD: 0.055 ± 0.031), which was not significantly different from that in HCs (*n* = 24; mean ± SD: 0.033 ± 0.0105; *p* = 0.1423) (Figure 6). 

### 3.6. Western Blot Analysis of PBMCs

The western blot analysis performed with proteins preparation obtained from PBMCs of sALS patients, PD patients, and HCs showed, as expected, ubiquitin conjugates overexpression in sALS and PD patients compared to in HCs. This results are consistent with accumulation of ubiquitinated proteins in PBMC cytoplasm of ALS and PD [40,45]. HSP90 also was overexpressed in PD controls and sALS patients but not in HCs (Figure 7a,b).

### 3.7. Genetic Analysis

The genetic analysis by NGS assays was performed to evaluate if VAPB clusters observed by IFA could be caused by VAPB gene mutation. Moreover, to best characterize our series of patients, we checked also for the presence of genetic mutation in the major ALS-related genes (*SOD-1*, *TARDPB*, *FUS*, and *C9orf72*). The data obtained revealed the presence of the A382T mutation of the *TARDBP* gene in three patients and pathogenic expansion in the *C9orf72* gene in one patient. The 20 remaining patients analyzed did not reveal the presence of mutation in any of the genes analyzed, with the VAPB gene included.

## 4. Discussion

The pathological mechanisms involved in ALS are complex and not completely understood. It is known that genetic factors play a role in motor neurons degeneration, but even in fALS cases, environmental factors influence the severity of this disease [46]. Protein aggregates are a constant feature of ALS [10], and their presence is responsible for cellular stress induction that leads to motor neurons apoptosis [47]. VAPB is involved in many cellular pathways essential for maintaining homeostasis and, when present in mutant form, tends to form intracellular insoluble aggregates, leading to ER impairment. Several studies, performed in human and transgenic mice, documented that VAPB aggregates can be found also when other ALS-causative genes, such *SOD1* [23], *TARDP* [24], and *C9orf7*2 [25], are mutated. Therefore, there are growing evidences that suggest VAPB involvement in ALS pathomechanisms [11,15,22]. Numerous experimental and ex vivo cell models have been proposed to study ALS pathological mechanisms [48], including PBMCs [27,28,29]. These cells in fact share more than 80% of transcriptomes with other tissues, including the central nervous system, and are easily isolable cells by noninvasive methods [26,29]. Thus, in the perspective of a new vision of ALS as a multisystemic pathology [30,31,32], we decided to use PBMCs to investigate the role of VAPB as a possible biomarker of the ALS pathology. 

To reach our goal, we first reproduced experimentally a pathologic model of VAPB-P56S mutation in HeLa cells, useful for subsequently analysis in PBMCs. In this regard, 24 patients with sALS and 14 patients with PDs as controls for neurodegenerative disorder and 24 HCs were recruited to deepen the role of VAPB in ALS. The data obtained in the VAPB-P56S-transfected HeLa cell model by performing IFA with an antihuman VAPB polyclonal antibody showed the presence of VAPB aggregates in close association with the ER, causing its impairment. It is known that these aggregates are a direct consequence of VAPB misfolding [11,13,38,49], and indeed, the western blot analysis revealed the increase of all the analyzed cellular stress markers. These data are consistent with those already reported in the literature about cellular models and postmortem spinal cord sections studies [11,39,47].

In line with the observations obtained in VAPB-P56S-transfected Hela cell model, the IFA performed with the same polyclonal antibody on PBMCs revealed, in all the 24 analyzed sALS patients, similar patterns of VAPB aggregates. Of relevance, HCs and PD controls did not show the presence of VAPB clusters evidenced in sALS patients, confirming the specificity of this alteration for ALS pathology. Therefore, to demonstrate the peculiar presence of such VAPB aggregates only in sALS patients, we analyzed also primary skin fibroblasts isolated from sALS, confirming our findings obtained in PBMCs of sALS. The identification of such patterns in fibroblasts and PBMCs of sALS patients led us to suppose that VAPB could be misfolded. For this reason, we performed an FCA using an anti-VAPB monoclonal antibody that specifically recognized the N-terminal domain of the protein (Figure 1), which guided the correct folding of VAPB. Interestingly, the FCA performed with the monoclonal antibody revealed a severe reduction of VAPB fluorescence signals in sALS patients but not in PD patients and HCs, suggesting that VAPB in sALS is post-transcriptionally modified/misfolded that renders the antibody to efficiently bind VAPB in sALS cases. However, in our series of sALS patients, the reduction of VAPB fluorescence signals did not seem to be associated with a variation in VAPB mRNA expression, and VAPB aggregates did not seem related to a genetic variation. In fact, by NGS analysis, no mutations in the VAPB-encoding gene were identified, and neither mutation was present in the *SOD1* and *FUS* genes. However, the *C9orf72* and *TARDBP* A382T mutations were identified in one and three patients, respectively. The frequencies of the mutations observed in this study did not differ significantly from the mutation frequencies reported in the Sardinian population [50,51]. The major limit of the study is the low size of samples presenting the mutant form of *TDP-43* and the *C9ORF* hexanucleotide expansion that did not allow appreciating any specific pathologic alteration related to VAPB ER-aggregates. 

It is noteworthy that, in this study, we report for the first time the presence of VAPB aggregates in PBMCs of sALS patients, finding out a consistent pattern of VAPB easily distinguishable from HCs and PD patients by IFA and FCAs. Until now, VAPB aggregates have been characterized and described in VAPB-P56S cellular models [11,13,15,18,38] in spinal cord sections of transgenic mice carrying VAPB-P56S mutation [23] and in cutaneous fibroblasts [52] and postmortem spinal cord sections of ALS-8 patients [22,53]. In all these studies, VAPB aggregates were described as insoluble and diffuse aggregates [21,23,52] that are in continuity with ER, causing its impairment [11,13,15,18,38]. These data are in line with our observations in sALS patients’ PBMCs and fibroblasts, on which we observed the presence of VAPB clusters similar to those reported in the studies mentioned above. However, our series of patients did not reveal the presence of mutations or alterations in VAPB gene expression. Therefore, the analysis and the interpretation of the data obtained by IFA and FCAs led us to hypothesize that VAPB aggregates may be a direct consequence of epigenetic modifications that result in an incorrect/different protein spatial conformation in sALS from those in HCs and PD patients. These results reinforce the idea that epigenetic studies can represent an attractive approach to giving a correct meaning to VAPB aggregates in pathogenic mechanisms of ALS [54].

In conclusions, our data are in good agreement with previous findings that VAPB is involved in ALS pathomechanisms and add a wedge to the understanding of ALS pathology. The study allowed us to find, in VAPB aggregates, an easy tool that permits distinguishing ALS from PD and could provide a possible platform for a much larger study including other neurodegenerative disorders. We identified the IFA and the FCAs, through the use of antihuman VAPB polyclonal and monoclonal antibodies, respectively, the best techniques to detect VAPB aggregates that could represent a novel diagnostic device for ALS. The importance of studying peripheral systems, such as PBMCs, supports a new vision of ALS pathomechanisms of a multisystemic disorder. In fact, these cells could serve as a noninvasive, simple, fast, cheap tool useful in clinical practice and research for studying ALS pathophysiology.

## Figures and Tables

**Figure 1 cells-09-00164-f001:**
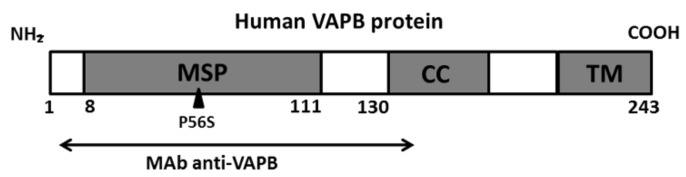
Schematic representation of the vesicle-associated membrane-protein-associated protein B (VAPB). The arrow indicates the recognition site for an anti-VAPB monoclonal antibody (2-130 aa) used for flow cytometry assays (FCAs). This site includes the N-terminal domain (8-111 aa) of the protein, called MSP that can be affected by P56S mutation.

**Figure 2 cells-09-00164-f002:**
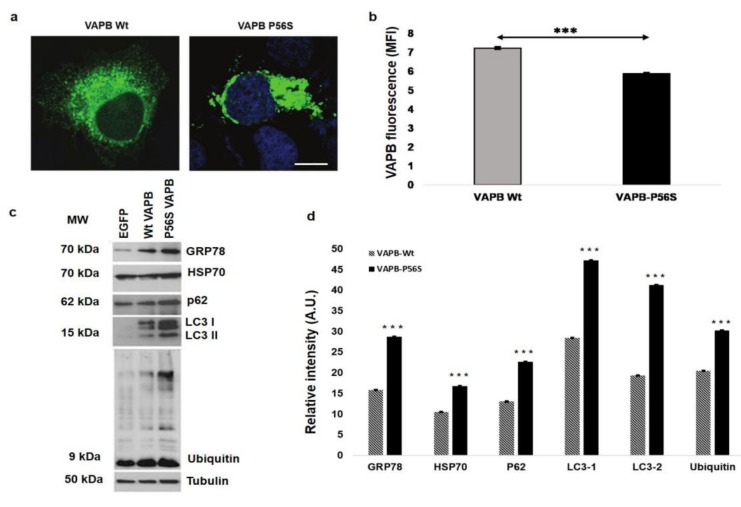
Representative immunofluorescence images performed with an antihuman VAPB polyclonal antibody in Hela cell lines. (**a**) VAPB (green) and nuclei (blue) in HeLa cells transfected with VAPB-Wt (left) and VAPB-P56S (right). Magnification is the same in both pictures (scale bar: 15 µm). The image shows VAPB aggregates and ER disorganization in VAPB-P56S-transfected HeLa cells caused by VAPB misfolding. (**b**) FCAs performed with an antihuman VAPB monoclonal antibody. The data, expressed as medium intensities of fluorescence (MFIs), show a statistically significant reduction of fluorescence signals in VAPB-P56S-transfected HeLa cells compared to in VAPB-Wt-transfected HeLa cells. (**c**) Representative western blot analysis showing the expression of GRP78, p62, and LC3 proteins and ubiquitin protein in VAPB-Wt- and VAPB-P56S-transfected cells. β-tubulin was used as a loading control. For each analysis, around 15 µg of protein were loaded. All western blot experiments were performed in triplicate. (**d**) Quantification of band intensities normalized with a β-tubulin value depicted in (c). Densitometry analysis was performed using the software ImageJ. Data are from one representative experiment out of three. *** *p* < 0.0005.

**Figure 3 cells-09-00164-f003:**
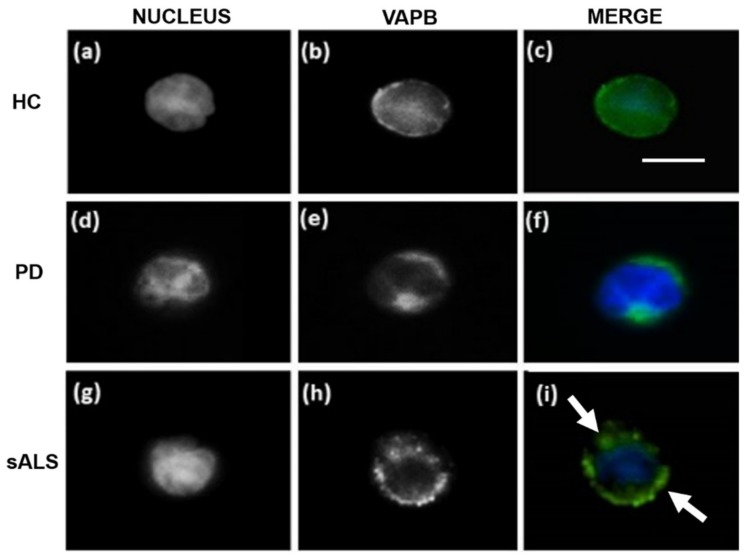
Representative confocal monochromatic nucleus (**a,d,g**), VAPB (**b,e,h**), and merge (**c,f,i**) images of immunofluorescence performed with an antihuman VAPB polyclonal antibody in PBMCs of HCs, PD patients, and sALS patients. VAPB is represented in green, and the nuclei are in blue. (**i**) The merge image shows the presence of VAPB ER-aggregates in sALS patients that is clearly different from those obtained in PD patients (**f**) and HCs (**c**). The arrows indicate VAPB ER-aggregates. Scale bars: 20 µm. Each sample were analyzed in 20 fields. For each subject, the number of lymphocytes showing VAPB aggregates was divided by the original total number of isolated cells (2 × 10^4^).

**Figure 4 cells-09-00164-f004:**
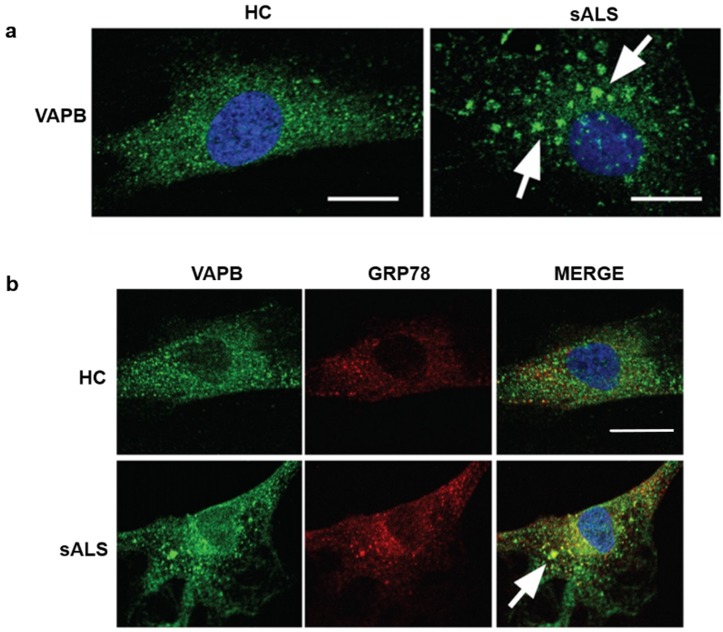
(**a**) Representative immunofluorescence image of primary skin fibroblasts performed with an antihuman VAPB polyclonal antibody. VAPB (green) and nuclei (blue) in fibroblasts were isolated from HCs (left) and sALS patients (right). The sALS patient fibroblasts staining showed globular accumulations of VAPB (indicated by arrows) compared to the control. (**b**) Image showing single staining of VAPB (green) (left), GRP78 (red) (middle), and merge (right). The merge evidenced the colocalization of GRP-78 with the VAPB accumulations (indicated by arrows) representing the ER distribution. Scale bars: 15 µm.

**Figure 5 cells-09-00164-f005:**
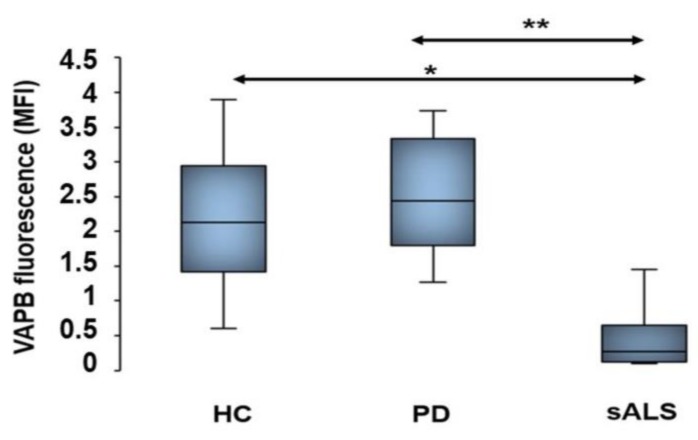
FCAs performed with a monoclonal antibody against VAPB. The graph shows a statistically significant decreased level of VAPB fluorescence detection in sALS patients compared to in PD patients and HCs. On the contrary, VAPB fluorescence signals in HCs and PD patients did not present significant differences. The data are expressed as MFIs in all patients and controls analyzed. * *p* < 0.05; ** *p* < 0.005.

**Figure 6 cells-09-00164-f006:**
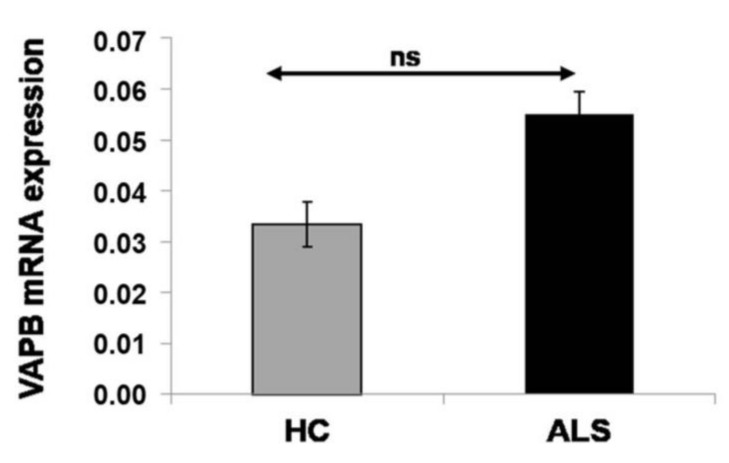
VAPB mRNA expression in PBMCs from HCs (*n* = 24) and ALS (*n* = 24) patients. mRNA levels were quantified by real-time PCR. mRNA values are normalized by that of GAPDH. Values represent mean ± SD. Each sample was examined in triplicate. VAPB mRNA levels of sALS patients were not significantly different from those detected in HCs.

**Figure 7 cells-09-00164-f007:**
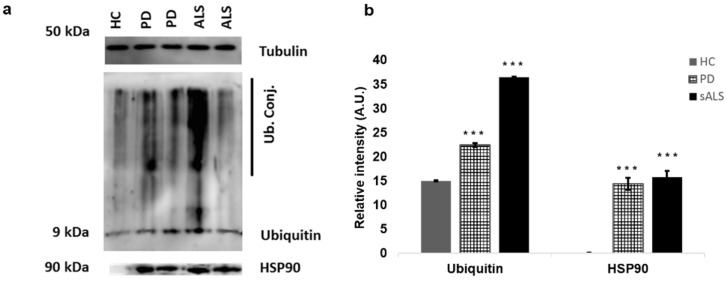
(**a**) Representative western blot data from one HC and two PD- and two sALS-affected patients. The image shows the overexpression of ubiquitin and HSP90 in sALS and PD patients compared to in the HCs. For each analysis, around 15 µg of protein were loaded. β-tubulin was used as a loading control. (**b**) Quantification of band intensities normalized with a β-tubulin value depicted in (a). Densitometry analysis was performed using the software Image. All western blot experiments were performed in triplicate. *** *p* < 0.0005.

**Table 1 cells-09-00164-t001:** Demographic and clinical characteristics of patients with amyotrophic lateral sclerosis (ALS) and with Parkinson’s disease (PD) and healthy controls (HCs).

	sALS	PD	HC
Number	24	14	24
Mean age ± SD	66.1 ± 6	71.3 ± 9.6	63.2 ± 5.8
Mean age at onset ± SD	59.7 ± 10.6	63 ± 11.9	
Males/Females ratio	16/8	8/6	16/8
Disease duration ± SD ^1^	4.6 ± 4.1	8.3 ± 4.8	-
Signs at onset: spinal	19 cases	-	-
Signs at onset: bulbar	5 cases	-	-
H&Y score, median ± SD	-	3 ± 1.2	-
ALSFRS-R, median ± SD	36 ± 5.1	-	-

^1^ Disease duration is expressed in year.

**Table 2 cells-09-00164-t002:** Primer pairs used for next-generation sequencing (NGS) assays.

Gene	Exon	Forward Primer Sequence	Reverse Primer Sequence
***TARDBP***	1–3	5′-3′ *GGGGAGGTCAGCTCCTATAC*	5′-3′ *TCATCTGTGTAACAGAAAGGACAGT*
	4–6	5′-3′ *CCACTGCATCCAGTTGAAACC*	5′-3′ *TGCCCACCCAGTGTAATGTC*
***FUS***	1–6	5′-3′ *TAGTCCTGCCGAGGAGAGAG*	5′-3′ *CTACACTGCGAGAAGAGCGG*
	7–15	5′-3′ *GTCAGACAAGGGGTGGTCAG*	5′-3′ *CTGGCAGGGGAAGTAAACCT*
***SOD***	1	5′-3′ *TTTCGGGGTTCTGGACGTTT*	5′-3′ *TAGGTTGTGCCAGCGTCTAC*
***VAPB***	1	5′-3′ *TGGCTGTCGTAGCTGGACTA*	5′-3′ *GTCCTTCCAGCACTGGGG*
	2	5′-3′ *GTGAGGTTTCAATGAGATGCCA*	5′-3′ *TGGGACCCAGCTTTCTGAT*
	3	5′-3′ *CAGCTCTGTCATGCGTGATTT*	5′-3′ *ACAGCCTTCTAGAAAGTCACAAAA*
	4–6	5′-3′ *TTTTGTTCAGTGGTGGGCCT*	5′-3′ *AAACATGGCATGGAGGGAGG*

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
