# Peer review of "VAPB ER-Aggregates, A Possible New Biomarker in ALS Pathology"

_cells, 2020, doi:10.3390/cells9010164_

Round 1

Reviewer 1 Report

The authors report the identification of VAPB aggregates as potential new non-invasive biomarker in peripheral blood mononuclear cells (PBMCs) of patients with sporadic form of ALS.

They also identified IFA and FCA as best techniques to detect VAPB aggregates in diagnostics for ALS through the use of anti-human VAPB polyclonal  and monoclonal antibodies, respectively.

The article is methodologycaly correct and well written.

My  concern is, because authors only compared two neurodegenerative diseases (ALS and PD),  that the following statement :

"The study allowed us to find, in VAPB aggregates, an easy tool that could permit to distinguish ALS from other neurodegenerative disorders." should be changed. 

Author Response

Point 1: My concern is, because authors only compared two neurodegenerative diseases (ALS and PD), that the following statement: "The study allowed us to find, in VAPB aggregates, an easy tool that could permit to distinguish ALS from other neurodegenerative disorders." should be changed.

Response 1: We are very grateful to the reviewer to raise this point. Please see in discussion section. 

Reviewer 2 Report

Review

VAPB ER-aggregates, a possible new biomarker in 2 ALS pathology

Maria Piera L Cadoni, et al

Missense mutations in VAPB cause familial ALS type 8, but the precise mechanisms underlying the pathophysiology of this disease are not yet clear. Furthermore, it remains to be investigated whether misfolded VAPB also plays a role in sporadic ALS.

In this study, authors employed human cell models to further investigate the role of VAPB in neurodegeneration. They found an abnormal VAPB staining pattern in PBMCs from patients with ALS, not found in subjects with Parkinson’s disease (PD) or healthy controls (HC). In addition, using flow cytometry, they were able to show significantly reduced VAPB fluoresce signal in PBMCs from subjects with ALS compared to PD and HC.

There are few studies focusing in VAPB, so the current manuscript adds novel information not only for the basic understanding of ALS, but also for clinical care of the disease (suggesting a novel potential diagnostic marker).

The text would benefit from language revision.

Additional comments

Authors should provide more details about the ALS cohort (age at onset, bulbar vs spinal, definite/probable/possible, etc). In table 1, disease duration is reported as 4.6±4.1, but the time unit is not shown. For typical ALS, a mean disease duration of 4.6 years would be quite unusual. Two antibodies against VAPB were employed (monoclonal and custom-made). Please emphasize the main differences between each of them (ie, the target epitopes of each one). How did authors evaluate the concentration and quality of the mRNA extracted from PBMCs? Please give more details in the methods section. Some comparisons included more than 2 groups. Did authors employ any adjustment for multiple comparisons for these analyses? Figure 1: Please provide the magnification of image 1a. Please include the SE bar on the top of the columns. I would recommend to include the blots of all proteins (including the reference beta-tubulin) in a single gel. It would be important to have more experimental details on the densitometric analyses of WB (Figure 1d) in the methods section. Did authors perform VAPB expression analyses in PBMC from the PD cohort? If so, please include the results in Figure 6. How did these IF findings compare to those reported in cells derived from patients with VAPB-related ALS? Please comment in the discussion section.

Author Response

Response to Reviewer 2 comments

Point 1: Authors should provide more details about the ALS cohort (age at onset, bulbar vs spinal, definite/probable/possible, etc).

Response 1: we revised the MS and we added all the information in table1 and in 2.3 methods section.

Point 2: In table 1, disease duration is reported as 4.6±4.1, but the time unit is not shown.

Response 2:  we regret the confusion  and we added the time unit of disease duration in table 1

Point 3: For typical ALS, a mean disease duration of 4.6 years would be quite unusual.

Response 3 : We are very grateful to the reviewer to raise this point. For “classical” ALS the median survival period following onset is usually 2-4 yr (Logroscino G, et al. J Neurol Neurosurg Psychiatry 2010). In the ALS cohort of our study a mean disease duration of 4.6 years was likely related to the high percentage of patients (70%) treated with permanent tracheostomy-intermittent positive-passive ventilation that allows long-term survival for patients with respiratory failure (Marchese S et al Respiratory Medicine 2008).

Point 4: Two antibodies against VAPB were employed (monoclonal and custom-made). Please emphasize the main differences between each of them (ie, the target epitopes of each one).

Response 4: we revised the MS and we added all the information in 2.7 methods section.

Point 5: How did authors evaluate the concentration and quality of the mRNA extracted from PBMCs? Please give more details in the methods section.

Response 5: we revised the MS and we added all the information in 2.9 methods section

Point 6: Some comparisons included more than 2 groups. Did authors employ any adjustment for multiple comparisons for these analyses?

Response 6: Visual inspection of boxes (Cytometry graph) suggested us to perform comparison between couples of independent data. Therefore, for our data there is no need to use multiple comparison tests.

Point 7: Figure 1: Please provide the magnification of image 1a. Please include the SE bar on the top of the columns. I would recommend to include the blots of all proteins (including the reference beta-tubulin) in a single gel.

Response 7: we revised the MS, please see results section.

Point 8: It would be important to have more experimental details on the densitometric analyses of WB (Figure 1d) in the methods section.

Response 8: we revised the MS, please see 2.5 methods section (Fig 2d).

Point 9: Did authors perform VAPB expression analyses in PBMC from the PD cohort? If so, please include the results in Figure 6.

Response 9: VAPB expression analyses in PBMC from the PD cohort was not performed.

Point 10: How did these IF findings compare to those reported in cells derived from patients with VAPB-related ALS? Please comment in the discussion section.

Response 10: We are very grateful to the reviewer to raise this point. Please see the discussion section.

Point 11: The text would benefit from language revision.

Response 11: the text has been revised by a native English speaking colleague.

This manuscript is a resubmission of an earlier submission. The following is a list of the peer review reports and author responses from that submission.

Round 1

Reviewer 1 Report

Summary:

The authors of the submitted manuscript entitled “VAPB ER-aggregates, a possible new biomarker in ALS pathology” described the analysis of Hela cell overexpression of vesicle-associated membrane protein-associated protein B (VAPB) and differences of the expression VAPB in PBMCs isolated from healthy controls, PD patient, and ALS patient. VAPB has been shown to be associated with forms of ALS, which makes it a potentially interesting biomarker for the disease. The authors isolated patient PBMCs to assess the expression and changes in VAPB between ALS and PD patients, as well as healthy controls. Overexpression of a mutated form of VAPB (P56S) in Hela cells increased cell stress, degradation, and autophagy markers compared to WT and control cells, indicating induced pathology in the presence of this mutant protein and supporting a role of VAPB in ALS-related cellular pathology. Since VAPB is involved in cellular, breakdown, transport and interaction mechanisms, the immunofluorescent visualization of increased fluorescent puncta in the mutant protein-expressing cells is reasonable. Ultimately, the study suggests that PBMCs could be used to assess VAPB as a marker for ALS. This is an interesting study with clinically-relevant implications; however, some issues described below should be addressed to clarify some of the points in the paper as well as to improve the quality of the manuscript.

Major:

1) The point mutations in VAPB are mentioned as an important factor in the dysregulation of this protein’s function. Also, in later experiments, such as immunolabeling and assessment of isolated cells, it is not clearly presented how the antibody utilized can detect the mutated forms of the protein in cells. Though the authors show a diagram of where the mutated region lies on the protein, and the proposed binding site of the antibody, the protein conformation may be different due to the mutation which could help distinguish between mutated and non-mutated forms.

2) Alternatively, the antibody may also recognize both non-mutated and mutated forms of the protein, which clouds the interpretation of the data using the antibody as well as the initially discussed idea of the importance in the mutation in detection and disease. As such, there is a disconnect in 1) what the role and importance of the mutation is in the quest for a biomarker of the disease, 2) what this means for pathologic progression, and 3) how the data in the paper bring all of this information together. I highly recommend the authors revise the manuscript to clearly describe at each experimental result in the study how the mutation is detectable and useful for the given assessment. Right now, it seems to be important at some points and less important in others.

3) Figures 2 and 3 do not seem to support each other, as fluorescent labeling of expressed VAPB seems far greater in ALS patient PBMCs in Figure 2,  yet in Figure 3, the quantitative data show VAPB expression is significantly less than PD patient and healthy control PBMCs. The authors should clarify this discrepancy with more detail, as the Discussion section provides some insight into the antibody differences in labeling (polyclonal vs. monoclonal) but this is difficult to understand in the context of the flow of the study.

4) Many of these issues relate back to major point #1 concerning the ability to detect different forms of the VAPB protein and relevance of doing so for the purpose of PBMCs VAPB expression and biomarker possibilities for ALS. Since this detection is critical for the studies central focus of using VAPB in PBMCs as a biomarker for ALS, getting into differences in monoclonal vs polyclonal antibodies is distracting from the way the story is presented. What is the mutation good for as a biomarker? What is the best way to detect it? How useful are PBMCs as characterizable cells from individuals for the assessment of VAPB and is the mutation important for this role for VAPB as a biomarker? I believe this is an important study and the rationale for the steps is understandable, but the presentation of the data and the importance of the different experiments in the context of what the study is trying to achieve is less readily apparent than would be necessary to communicate the value of the protein in PBMCs as a biomarker. This could be streamlined and the major focal points of the study could be presented concerning VAPB in PBMCs as a biomarker, with issues that influence how to detect the protein and its mutated form, and what this means in the context of the major goal of the study could be provided as Discussion points and Supplemental data.

Minor:

There are various typographical errors and poorly structured sentences that indicate this paper needs to be proofread and edited by someone with a stronger ability in writing English. This also affects the flow and comprehension of the studies points and Discussion. I highly recommend this be revised with the major points considered above by an individual with experience in writing technical and scientific language with greater proficiency in English. It is not terribly written, but it suffers enough that it is apparent and disrupts the quality of the paper.

Reviewer 2 Report

The work presented is original and interesting in the field of ALS mechanisms and biomarkers. However, the study is still preliminary.

Major points:

1. Line 79: give more detail on how the plasmid was obtained.

2. Line 96: give more detail on the anti-coagulant used.

3. Check the methodologies for statistical analysis. E.g., did you check for normality? Average and SD are parametric whereas Mann-Whitney test is non-parametric.

4. Figures legends: the number of replicates is missing. In addition, for immunofluorescence the number of cells analysed is missing. For the immunoblotting the amount of loaded protein is missing.

5. Figure 1. immunoblotting for VAPB (with monoclonal and polyclonal antibodies)? Figure 1a: colocalization with markers of the endomembrane system should be done; at least colocalization with an ER marker (e.g., calnexin) is required. Figure 1b: loading control?

6. Figure 6: unfocused. Colocalization with markers of the endomembrane system should be done.

7. Figure 3: was polyclonal antibody also used to monitor VAPB expression?

8. Figure 4: were all samples analysed?  Were the samples shown from individual patients? In this case the two ALS lanes correspond to patients with or without the identified mutations?

9. Figure 5: should appear in material and methods.

Minor points:

1. Line 41: "recently" should be removed

2. Line 57: "non-invasive" should be corrected.

3. Line 59: reference?

4. Last paragraph of introduction contains text that would be more suited for discussion.

5. Line 118: "loading control" instead of "housekeeping".

6. Definition of all abbreviations.